# Impact of Sample Preparation Methods on Single-Cell X-ray Microscopy and Light Elemental Analysis Evaluated by Combined Low Energy X-ray Fluorescence, STXM and AFM

**DOI:** 10.3390/molecules28041992

**Published:** 2023-02-20

**Authors:** Lucia Merolle, Lorella Pascolo, Luisa Zupin, Pietro Parisse, Valentina Bonanni, Gianluca Gariani, Sasa Kenig, Diana E. Bedolla, Sergio Crovella, Giuseppe Ricci, Stefano Iotti, Emil Malucelli, George Kourousias, Alessandra Gianoncelli

**Affiliations:** 1AUSL-IRCCS di Reggio Emilia, Transfusion Medicine Unit, 42123 Reggio Emilia, Italy; 2IRCCS Burlo Garofolo, Institute for Maternal and Child Health, 34137 Trieste, Italy; 3Consiglio Nazionale delle Ricerche, Istituto Officina dei Materiali, 34149 Trieste, Italy; 4Elettra—Sincrotrone Trieste S.C.p.A., Basovizza, 34149 Trieste, Italy; 5Faculty of Health Sciences, University of Primorska, Polje 42, 6310 Izola, Slovenia; 6Area Science Park, Padriciano 99, 34149 Trieste, Italy; 7Biological Science Program, Department of Biological and Environmental Science, College of Arts and Sciences, Qatar University, Doha 2713, Qatar; 8Department of Medical, Surgical, and Health Sciences, University of Trieste, 34149 Trieste, Italy; 9Department of Pharmacy and Biotechnology, University of Bologna, 40126 Bologna, Italy; 10National Institute of Biostructures and Biosystems, 00136 Rome, Italy

**Keywords:** XRF, single cells, fixation methods, AFM

## Abstract

Background: Although X-ray fluorescence microscopy is becoming a widely used technique for single-cell analysis, sample preparation for this microscopy remains one of the main challenges in obtaining optimal conditions for the measurements in the X-ray regime. The information available to researchers on sample treatment is inadequate and unclear, sometimes leading to wasted time and jeopardizing the experiment’s success. Many cell fixation methods have been described, but none of them have been systematically tested and declared the most suitable for synchrotron X-ray microscopy. Methods: The HEC-1-A endometrial cells, human spermatozoa, and human embryonic kidney (HEK-293) cells were fixed with organic solvents and cross-linking methods: 70% ethanol, 3.7%, and 2% paraformaldehyde; in addition, HEK-293 cells were subjected to methanol/ C_3_H_6_O treatment and cryofixation. Fixation methods were compared by coupling low-energy X-ray fluorescence with scanning transmission X-ray microscopy and atomic force microscopy. Results: Organic solvents lead to greater dehydration of cells, which has the most significant effect on the distribution and depletion of diffusion elements. Paraformaldehyde provides robust and reproducible data. Finally, the cryofixed cells provide the best morphology and element content results. Conclusion: Although cryofixation seems to be the most appropriate method as it allows for keeping cells closer to physiological conditions, it has some technical limitations. Paraformaldehyde, when used at the average concentration of 3.7%, is also an excellent alternative for X-ray microscopy.

## 1. Introduction

X-ray fluorescence (XRF) microscopy is a valuable technique for studying biological samples [1] because it provides the distribution of elements in tissues and cells non-destructively, without the need for stains or fluorescent probes. Mapping the distribution and content of endogenous and exogenous elements and metals is pivotal, especially for those elements with great biological importance, named also as “life elements” [2]. In addition, the ability to map samples at the nanometre scale provides the opportunity to correlate elemental distribution with cell morphology and potentially investigate the physio-pathological state of the cell or explore relevant metabolic pathways [3,4,5].

Single-cell analysis is particularly attractive to researchers from life sciences, biology, and biomedicine, as it offers the opportunity to better understand cellular diversity and specificity. Indeed, multicellular organisms are composed of different tissues and cells, and conventional cell-based assays mainly represent the average response of a mixed population of cells [6,7,8]. As previously reported, elemental analysis on a single cell provides unique complementary information to large population analyses [7]. X-ray microscopy at the single-cell level allows for a detailed analysis of the representative elements of a particular cell state [6] or highlights morphometric features often related to cell function and activity [9]. In this scenario, proper sample preparation is the first and most important step to enable proper XRF analyses [10].

In the experimental setup for XRF Microscopy (XRFM), especially in vacuum conditions, for instance when dealing with soft X-rays, the biological constituents of samples must be fixed before being analysed. Membranes made of silicon nitride (Si_3_N_4_) are often preferred as sample carriers for cells, as they offer high transparency and minimal X-ray absorption for X-ray microscopy and are a very suitable surface for cell attachment.

Therefore, the crucial step to obtaining reliable and accurate X-ray data on concentrations and distributions of chemical elements in the samples lies in the correct handling and fixation methods [10]. The latter is chosen depending on the additional processing steps and final analyses planned before or after XRFM. The major challenge is to achieve a state as close as possible to the physiological condition of the cell.

Most commonly used cell fixation protocols have been adopted and adapted from immunocytochemistry; therefore, many fixation methods are described in the literature.

The most commonly used and readily available fixation methods use chemical-based fixatives that block autodegradation processes, increase stability, and preserve morphology [11]. Once applied to a biological sample, they form cross-links between amino acids, proteins, and lipids. Chemical fixation allows for better preservation of morphology than coagulating fixatives (e.g., solvents and acetic acid) generally used for light microscopy.

Aldehyde-based fixation with paraformaldehyde or formaldehyde preserves cell structure with less loss of biochemical information than fixation with organic solvents [12]; however, it is suboptimal for preserving most elements and immobilizing proteins for further analysis. Many small molecules (such as ions) or macromolecules (such as carbohydrates, lipids, and nucleic acids) are not efficiently cross-linked, resulting in their extraction, replacement, or loss [9]. Fixation with ethanol leads to dehydration and precipitation of proteins, resulting in alteration of cell membrane structure, cytoplasmic organelles, and nuclear content [12]. In addition, chemical fixation with organic solvents and cross-linking reagents can lead to denaturation of the protein antigens, which can be problematic for downstream antibody labelling techniques [13].

Some studies have attempted to assess the extent and impact of sample preparation for XRF, especially in the hard X-ray range, to map elements heavier than phosphorus [14]. Recently, a comprehensive study by Perrin et al. [15] showed a difference in the elemental distribution and concentration of several elements, up to Na and Mg, with different preparation protocols. They suggested frozen fixation as the optimal sample preparation for μPIXE (micro Proton Induced X-ray Emission) and μXRF (micro XRF), without an extensive investigation of the cells’ morphology. Similarly, Jin et al. [16] conducted a systematic study comparing elemental maps of frozen cells with chemically fixed cells, also considering different wash buffers for rinsing the cells. They claim that plunge-frozen cells followed by freeze-drying and plunge-frozen cells measured in cryo-conditions are preferable for quantitative elemental evaluations, as they found significant loss of K and Cl in chemically fixed cells, along with increased content and altered distribution of Ca, possible loss of P, and possible redistribution of Zn [16]. Although cryofixation is considered by many the ideal currently available method for cell imaging [10], this preparation is limited by the need for a cryosystem in the preparation facility and by a potential rapid degradation of fixative-free samples. In cryofixation, the sample must be frozen fast enough for the water to change from its normal liquid state to its solid state (vitrification) without an ice crystal phase in between. Ice has a larger volume than water and can therefore cause deformation of the biological ultrastructure. While a variety of samples are suitable for rapid freezing, only a small volume of the sample will produce amorphous (ice-crystal free) ice. Viruses, proteins, organelles, and cells can be quickly iced; however, their volume makes tissues and multicellular organisms much more challenging to be properly cryofixated. Some experimental set-ups enable investigating frozen cells in a frozen hydrated state, under cooling [16]. Even if this condition allows for better sample preservation and reduces radiation damage effects, it requires a complex cryo-stage and cryo-transfer of the samples, which are not available in all facilities, making the experiments much more difficult starting from the sample preparation step till the final analysis [17].

Finally, sample preparation seems to influence the extent of radiation damage caused by soft X-rays during analyses [18,19,20]; therefore, this effect should also be taken into consideration during the selection of the ideal sample preparation.

In summary, the information available to researchers on sample treatment is still insufficient and sometimes ambiguous, leading to the uncertain success of the experiment.

In this study, we combined XRFM with atomic force microscopy (AFM) to evaluate how different fixatives can alter and influence the content and distribution of light chemical elements, diffusible ions, and cell morphology. The comparison will provide helpful information for synchrotron scientists to determine the most appropriate protocol for a particular study or cell model.

As cell models, we have chosen (i) the human HEC-1-A adenocarcinoma endometrial cell line as a model of an adherent immortalised tumour cell line, (ii) human spermatozoa as an example of primary suspension cells, and (iii) the immortalised human embryonic 293 cell line (HEK-293), which are genetically identical cells, thus ensuring repeatability of the experiments. As fixation methods, we have selected (i) 70% ethanol (EtOH), (ii) paraformaldehyde (PFA) 2% for 2′, and (iii) PFA 3.7% for 20′ for all the three cell lines. Additionally, for the HEK-293 cell line, two additional fixation methods were added: methanol/C_3_H_6_O (Acetone) and cryofixation.

The advantages and limitations of each preparation method were investigated using low-energy X-ray fluorescence (LEXRF), in conjunction with scanning transmission X-ray microscopy (STXM) and AFM. The advantage of AFM lies in its ability to non-destructively study the morphology of biological samples at the cellular and subcellular level with a spatial resolution in the nanometre range. Therefore, it is an ideal approach to assessing volume and thickness differences in biological samples. In this work, it was used prior to X-ray analyses.

## 2. Results

As shown in Figure 1, three different fixation procedures were applied to all samples selected for this study; HEC-1-A endometrial cells, spermatozoa, and the HEK-293 cell line were prepared with three different protocols, i.e., 70% ethanol (EtOH), paraformaldehyde (PFA) 2% for 2′, and PFA 3.7% for 20′. For the last cell line, i.e., HEK-293, two additional fixation methods were tested: methanol/C_3_H_6_O and cryofixation. AFM measurements with a spatial resolution of 200 nm were first performed on the different selected cells, which were then analysed by XRFM and STXM. For XRFM, the cells were analysed at 1.5 keV to determine the intracellular distribution and XRF intensities of essential life elements such as Na, Mg, and O; at the same time, absorption and differential phase contrast images were acquired in STXM mode. While XRFM allows for elemental distribution inspection, STXM imaging provides morphological information; absorption images are sensitive to different sample thickness and/or density, and phase contrast images highlight borders or structures.

The XRF results were compared with and complemented by the AFM images.

### 2.1. HEC-1-A Cells

#### 2.1.1. XRFM Analysis

The XRF elemental maps of selected HEC-1-A endometrial cells fixed with the three different protocols, i.e., 70% ethanol (EtOH), paraformaldehyde (PFA) 2% for 2′, and PFA 3.7% for 20′, are shown in Figure 2. Absorption and phase contrast images (STXM) show no significant change in cell morphology with PFA 3.7% for 20′ and PFA 2% for 2′ protocols. At the same time, a less defined membrane edge is observed with ethanol fixation. Indeed, in the differential phase contrast image (PhC) of Figure 2a, the cells’ borders appear more blurred or less sharp for EtOH treatment than in the corresponding PFA 3.7% or PFA 2% fixations (PhC in Figure 2b,c). In Figure 2d, we show the average Na, O, and Mg values extracted from all the XRF maps collected for the three protocols. Compared to the standard PFA of 3.7% for 20′, fixation with PFA of 2% for 2′ can result in lower light element content, especially Na and, to a lesser extent, Mg. Fixation with 70% ethanol retains the highest Na levels but results in a deficiency of Mg (Figure 2d). It is worth noting that even if we did not quantify the elements’ amount, average XRF counts of oxygen can be selected to compare the different fixation methods in terms of precision index, as shown in Figure 2e. The coefficient of variation was determined to assess the extent of variability relative to the population mean, as it is widely used to express the precision and repeatability of an assay; a higher coefficient of variation (CV) reflects a large dispersion around the mean. Although the mean fluorescence counts (MFC) are not statistically different when considering the uncertainties, PFA 2% has the highest CV (18.37%) compared to EtOH and PFA 3.7%, for which CV was 5.95% and 9.20%, respectively (see Appendix A). All in all, fixation with PFA 3.7% for 20′ seems to preserve the internal cellular structure and thereby the Na and Mg content in the cells more efficiently (Figure 2d).

#### 2.1.2. AFM 

AFM microscopy was performed prior to XRF measurements on all cell lines to investigate the effects of the fixation procedure on cell morphology and volume changes. The AFM images of HEC-1-A endometrial cells fixed by the different methods and grown on Si_3_N_4_ membranes are shown in Figure 3. The three representative cell groups selected in Figure 2 are shown in the first panels of Figure 3a (EtOH), Figure 3b (PFA 2%), and Figure 3c (PFA 3.7%), together with other cells prepared with the same fixation method not shown in Figure 2 for spatial constraints. The AFM images confirm that there is no significant difference in cell morphology between the PFA 2% and PFA 3.7% protocols. Cells fixed with ethanol (Figure 3a) appear emptier and less compact, with the membrane edges tending to become less sharp and less defined. This is in agreement with what has been previously highlighted by comparing the differential phase contrast images of Figure 2.

Cells in PFA 3.7% have a higher mean volume (264.1 ± 212.3) overall compared to PFA 2% (196.0 ± 104.8) and EtOH 70% (163.3 ± 104.8); although the three values are comparable within their uncertainties, there is a progressive reduction of the cell volume from PFA 3.7% to EtOH.

### 2.2. Spermatozoa

#### 2.2.1. XRFM Analysis

Semen was collected from three healthy volunteer donors. Fixation with PFA 2% (Figure 4b) and 3.7% (Figure 4c) does not alter the morphological phenotype of the spermatozoa, which is characterised by a typically oval head and regular shape of the neck, midpiece, and tail. In contrast, when fixed with ethanol 70%, they appear dehydrated, and the tails are strongly curved (Figure 4a). Figure 4d shows the average counts of O, Na, and Mg on the heads of the spermatozoa for each of the preparations (values are normalised for the number of analysed spermatozoa). Fixation with PFA 3.7% can maintain a higher amount of Na and Mg (Figure 4c) compared to ethanol (Figure 4a) and PFA 2% (Figure 4b) fixations. Due to the large variability in the sample under study, ethanol and PFA 2% do not really show statistically significant differences compared with each other. Figure 4e shows the distribution of sperm oxygen levels; in this case, the coefficient of variation for PFA 3.7% (17.14%) is higher than for EtOH 70% (10.51%) and PFA 2% (7.1%), respectively. Spermatozoa XRF data show a trend which can be considered in agreement with that shown for HEC-1-A cells concerning PFA 3.7% preparation. However, a lower reproducibility level is observed, with high values of uncertainties that can be attributed to the peculiar feature of this cell type; indeed, we found a high degree of variability between different donors and sometimes between cells from the same donor (see Appendix A).

#### 2.2.2. AFM 

Figure 5 shows the AFM results for the same spermatozoa groups shown in Figure 4. The mean values of cell volume are comparable for the PFA 3.7% and PFA 2% fixatives, while a significantly lower volume is calculated for the cells fixed with ethanol (Figure 5d).

### 2.3. HEK-293

#### 2.3.1. XRFM Analysis

The HEK-293 cell line was also fixed with PFA 2%, 3.7%, and EtOH. Moreover, additional fixation methods were tested on the HEK-293 cell line, as shown in Figure 1 and described in Section 4. All XRF data are summarised in Figure 6. The PFA 3.7% protocol again proves to be a valid preparation method that appropriately preserves all metals and “living elements”. Although the ethanol method does not seem effective in preserving the Mg content in the cells, the Na XRF intensity is well-preserved (Figure 6a). Interestingly, the methanol/C_3_H_6_O and cryofixation methods show the highest Mg XRF counts in the cells, followed by PFA 3.7%, while PFA 2% and ethanol favour the loss of Mg XRF intensity. As for Na, methanol/C_3_H_6_O has the highest level, followed by PFA 3.7%, while cryofixation, EtOH, and PFA 2% show acceptable levels, although lower (Figure 6f). The washing processes increase the depletion of Na and Mg (see Appendix A). It is worth noting that the overall average O XRF counts are less affected by the preparation compared to Na and Mg and can be considered as a reference for comparing the possible Na and Mg depletions among the different fixation methods. From the absorption images and elemental distribution maps, it is possible to appreciate how fixation with ethanol and 3.7% PFA give similar results, while the absorption image of the cell fixed with 2% PFA is blurred, indicating partial degradation of the sample. Although the Na and Mg XRF intensity seems to be less affected by methanol/C_3_H_6_O and cryofixation, Figure 6d shows how methanol/ C_3_H_6_O fixation increases the precipitation of Na aggregates on the cells, which is visible in both the absorption and phase contrast images and even more so in the Na-XRF map; this is probably due to a reaction between the solvent and the PBS buffer used to wash the cells. This artefact accounts for the highest value of Na for methanol/C_3_H_6_O fixed cells. Cells that have been fixed with methanol/ C_3_H_6_O and then washed do not show the same Na clusters (Appendix A), but the cells appear to be “depleted”, and the total amounts of O, Na, and Mg decrease significantly. Similar considerations apply when comparing ethanol fixation with the same fixation followed by subsequent washing (see Appendix A). The PFA 2% protocol achieves similar results to the ethanol protocol concerning the O, Na, and Mg XRF counts, but the absorption and phase contrast images clearly show impaired cell morphology. The coefficient of variation of oxygen, used as the dispersion parameter of the XRF measurement, changes between the different methods, with very high values for the cryofixed cells and the ethanol-treated cells. At the same time, the other protocols give relatively low values of CV (see Appendix A).

#### 2.3.2. AFM 

Interestingly, AFM images of methanol/C_3_H_6_O preparation (Figure 7d) show that the morphology of HEK-293 cells is significantly altered, with submicrometric features appearing like clumps on their surface. As mentioned earlier, according to the XRF maps, these features appear to be closely associated with Na nanoclusters. When prepared with ethanol (Figure 7a), the morphology is better preserved, although the cells appear somewhat depleted. This is even more pronounced in the cells fixed with PFA 2% (Figure 7b), as their overall height appears reduced compared to methanol and ethanol fixation; moreover, the cells prepared with PFA 2% have visible nanometric holes on their surface, clearly indicating some stress. In contrast, cells prepared with PFA 3.7% (Figure 7c) appear more filled, as evidenced by their height, and have a smoother surface; overall, they appear healthier and with a better-preserved morphology. AFM images with corresponding surface profiles collected on a selection of (a) 1:1 MeOH/C_3_H_6_O washed cells and of (b) 70% ethanol washed cells are reported in Appendix A.

## 3. Discussion

XRF analysis is a powerful, non-destructive, analytical technique with wide application in biomedical science, allowing the chemical characterisation, with good detection limit and high spatial resolution, of trace elements that may play an essential role in cellular biochemical processes. We have recently presented some important results on cells derived from reproductive tissues, observing Na, Cu, and Mg distribution in spermatozoa [21] and ovarian tissues [22]. In another work, we also demonstrated that it is possible to merge compositional and morphological information to quantitatively derive the element concentration by combining XRFM with AFM and STXM analyses [23]. The faithfulness of XRF measurements is highly dependent on the proper preparation of the samples, which is indeed a critical step in obtaining the appropriate information; it requires careful attention in order to preserve the cellular morphology and the elemental localisation inside the cells.

The present study illustrates the effect of several cell fixation methods in single-cell analyses on the localisation and intracellular levels of light elements by synchrotron radiation XRF. As already reported, the most affected elements are the most diffusible ones, namely the light ones, such as Na, Mg, K, Cl, and Ca [15,16]. However, in the available literature, there is a lack of a comprehensive evaluation of the morphological changes correlated to elemental preservation or possible losses after fixation. 

We focused on analysing the impact of preparation techniques on the determination of “life elements” O, Na, and Mg inside the cells, using the TwinMic soft X-ray microscope, which also allows for morphological investigations through high-resolution (sub-micrometric length scale) absorption and phase contrast images [23]. Even if the concentration of intracellular elements inside the cells cannot be represented by XRF counts nor MFC parameter alone, the present study aims at illustrating the effect of different fixation methods on cell morphology and elemental distribution. Oxygen XRF counts can be used as a reference to compare Na and Mg changes, while AFM microscopy performed prior to X-ray analyses indeed enabled us to examine at nanometric resolution the changes in cell volumes and morphology induced by the different fixation methods.

In order to provide a more systematic and comprehensive understanding, we decided to compare different fixation protocols on three cell models. Cell fixation methods were chosen accordingly to the following motives: (i) cross-linker, (ii) organic solvent, and (iii) frozen methods. Usually, chemical fixation preserves structures in a state (both chemically and structurally) as close to living tissue as possible by stabilising the proteins, nucleic acids, and other macromolecules. For instance, PFA 3.7% fixation is among the most used for X-ray microscopy and immune- and histochemistry analyses, having the ability to permeabilise cells, in addition to a quick fixation period. Different chemical fixatives are characterised by different permeabilisation grades, reflecting anomalous elemental diffusion from the cells. For this reason, we tested a few examples of chemical fixatives: PFA 3.7%, PFA 2%, ethanol 70%, and 1:1 methanol/ C_3_H_6_O. While the first three methods were applied to all cell models, methanol/ C_3_H_6_O and cryofixation were used only for the HEK-293 cells, which are highly reproducible, thus allowing for a more comprehensive analysis. 

Our results confirm that the PFA 3.7% protocol is an excellent compromise to preserve both morphological features, including cell volume, and light element content to a reasonable extent. The findings align with Perrin et al. [15], confirming that Na XRF intensity is well-preserved in PFA 3.7% treated cells and even better for the MeOH/C_3_H_6_O protocol. However, when investigating at a submicrometric length scale, our results demonstrate that the high Na levels after methanol fixation are due to nanometric Na agglomerates or precipitates onto the cells (Appendix A), thus excluding this fixative since it introduces evident artefacts. Moreover, our results evidence that the fixations with organic solvents greatly affect the morphology or the volume, probably due to their dehydrating effect.

While we still observe significant Mg levels for both PFA 3.7% and MeOH/C_3_H_6_O fixations, for Perrin et al. [15] its elemental content seems to be reduced below the detection limits in both cases. PFA 2% does not show any advantage compared to PFA 3.7%, as the cells treated with this method usually present a lower cell volume and a lower XRF counts of Mg. Furthermore, we compared those results with ethanol fixation, which is a frequently used protocol, highlighting good preservation of Na levels, but not of Mg and cellular volume. All the observations mentioned above concerning PFA 3.7%, PFA 2%, and ethanol effects on elemental XRF intensities, and thus on elemental content, are substantially confirmed in all cell models used in this study (Figure 2, Figure 3, Figure 4, Figure 5, Figure 6 and Figure 7), providing a good consistency to our results. However, AFM analyses revealed that the way PFA 2% affects the three cell models differs in terms of cell volume, with HEC-1-A endometrial cells the least affected (Figure 3). Spermatozoa cells show the highest variability in cell volume modifications when fixed with PFA 2%, suggesting that the cell dimension, which in this case is highly variable, may affect the final deformation. The maximal effect for the HEK-293 embryonic line is likely due to their undifferentiated phenotype, as seen in Figure 7b, where the PFA 2% fixed cells seem to collapse and cover a wider surface. In addition, they present irregular cellular profiles. Hence, too mild of a treatment with PFA is insufficient to cross-link the structural cell component, thus resulting in rapid degradation.

Our data with HEK-293 cells suggest cryofixation as the best method, as it ensures the highest cell volume, very likely the closest to the physiological one. Moreover, in agreement with Perrin et al. [15], it maintains a high level of Mg while inducing a moderate loss of Na (Figure 6e). However, cryofixation with freeze-drying is a sensitive step because biological samples usually contain a high percentage of water. If not properly performed, it can lead to elemental redistribution, loss of elements, shrinking of the specimens, and severe morphology distortion [24].

Cryofixation provides the most truthful representation of the cellular biological structure; however, a limit is the lower repeatability compared to other protocols, as shown by the measurement of the oxygen coefficient of variation (Figure 6g). This is probably due to an intrinsic technical difficulty of operating in a standardised way [25]. To summarise, PFA fixation at 3.7% is the recommended protocol in the absence of cryofixation, even though our results and other ones from literature [15,16,26] suggest that all chemical fixatives affect some aspects of the measurement.

Finally, attention must be paid to water washing after fixation since our results demonstrate that it has some influence on elemental XRF counts, as organic fixatives are known to perforate cell membranes, inducing a release of elemental content outside the cell.

The main analytical constraints of the methods under study are summarised in Table 1 according to our study and the literature [15,16,26].

To note, the study was performed on a limited population of cells due to two main factors: (i) access to a synchrotron facility is granted only for scientific merit, assessed by a peer review process carried out every 6 months; (ii) low-energy XRF is not an efficient phenomenon, as below 2 keV the Auger effect is the dominant process, which results in a low florescence yield and thus long measurement times. Despite these intrinsic limitations, the reported results still provide very useful information to the X-ray microscopy and XRF community, and further complement and confirm the results available in literature, which suffer as well from similar constraints [15,16].

## 4. Materials and Methods

Silicon nitride membranes Si_3_N_4_ (100 nm thick; Silson, Southam, UK) were used as support for both XRFM and AFM analyses for all the specimens under study. Si_3_N_4_ were previously sterilised under a laminar flow hood with EtOH 70%, which was left to evaporate overnight in a petri dish.

### 4.1. Cellular Models

#### 4.1.1. HEC-1-A Endometrial Cells

HEC-1-A endometrial cells from human adenocarcinoma (ATCC HTB-112) were cultured in McCoy’s 5a Medium supplemented with 10% foetal bovine serum, 100 units/mL penicillin, and 100 μg/mL streptomycin (Sigma Aldrich, Milan, Italy). For the experiments 50.000 cells were seeded in 12 multiwell plates containing silicon nitride membranes (100 nm thick; Silson, Southam, UK). After 24 h, the cells were fixed with three different protocols:Paraformaldehyde (PFA) 3.7% in phosphate buffer saline (PBS) for 20 min at room temperaturePFA 2% for 2 min at room temperature70% ice-cold ethanol for 3 min at −20 °C

Then the membranes were washed twice in PBS and twice in distilled water and air-dried.

#### 4.1.2. Spermatozoa

Three fresh semen samples were collected from European Caucasian male volunteer donors enrolled at the assisted procreation unit (Institute for Maternal and Child Health IRCCS Burlo Garofolo, Trieste, Italy). All participants signed an informed consent. All experimental procedures were conducted according to the ethical standards of the Declaration of Helsinki (7th version 2013); the study was approved by the regional (FVG) Ethics Committee (Comitato Etico Unico Regionale FVG) (CEUR-2019-Os-191). The specimens were collected via masturbation after 2–7 days of sexual abstinence. After liquefaction, the samples were observed under an optic microscope and they presented normal parameters in terms of concentration and motility (according to the World Health Organization guidelines) [27].

The semen from each donor were divided in 3 groups and fixed with:PFA 3.7% for 20 min at room temperaturePFA 2% for 2 min at room temperature70% ice-cold ethanol for 3 min at −20 °C

Then the samples were washed twice in physiological solution (0.9% NaCl in water), and after centrifugation at 300× *g* for 10 min they were resuspended in physiological solution, deposited on 100 nm thick Si_3_N_4_ membranes, and air-dried. The membranes were briefly dipped twice in distilled water and air-dried.

#### 4.1.3. HEK-293 Cell Line

Human Embryonic Kidney 293 cells (HEK-293) were cultured in Dulbecco’s Modified Eagle Medium (DMEM; Biochrom, Berlin, Germany) supplemented with 10% foetal bovine serum (FBS; Biochrom, Berlin, Germany) on silicon nitride membranes (100 nm thick) at 37 °C and 5% CO_2_ atmosphere. Cells were left to attach overnight and washed with phosphate buffer saline (PBS; Euroclone, Milan, Italy). Subsequently, cells were subjected to the following fixation methods:PFA 3.7% for 20 min at room temperaturePFA 2% for 2 min at room temperature70% ice-cold ethanol for 3 min at −20 °C1:1 MeOH/C_3_H_6_O for 3 min at −20 °C

Then all membranes were washed twice in PBS and twice in distilled water and air-dried.
5.Cryofixation: cells were washed with ammonium acetate buffer solution 100 mM pH 7.4 prepared with high purity water (Fisher Scientific™ Accu100 Ultrapure Water System), the excess washing buffer was removed, and then Si_3_N_4_ membranes were rinsed in ethane (−160 °C) by using a home-made machine; cells were subsequently freeze-dried by leaving the specimens overnight in a controlled pressure system, allowing the ice to sublimate and slowly reach atmospheric pressure and temperature.

All reagents used for fixation are highly pure ones bought from Sigma Aldrich—Merck (Milano, Italy) (PFA 158127, Ethanol 32221, PBS P4417). The PFA was also filtered with a 0.2 micron filter prior to use to eliminate aggregate samples.

### 4.2. AFM Microscopy 

The AFM measurements were performed at the NanoInnovation Laboratory located at Elettra—Sincrotrone Trieste (Trieste, Italy). Suitable cells, prepared as described in Section 4.1, were selected through visible light microscopy and then mapped with AFM microscopy prior to XRF analysis. This was done in order to avoid any possible artefacts caused by radiation damage induced by soft X-rays. AFM micrographs were acquired in contact mode in air with an XE100 (Park Instruments) or a MFP3D (Asylume Research/Oxford Instruments) instrument. For the acquisition of images, we used soft cantilevers (Mikromasch CSC38, radius of curvature <10 nm, spring constant 0.006 N/m) at a 0.1–0.2 Hz scanning rate and a 80–160 nm pixel resolution. Images were analysed with Gwyddion software [28]; volume extraction was carried out by flattening the image to remove background, selecting the cells by a height-threshold based mask, and measuring the zero-basis volume of the single cells. Graphs of AFM trace profiles were obtained using the Igor Pro software (Wavemetrics, Lake Oswego, OR, USA).

### 4.3. XRF Microscopy 

The XRF measurements were performed at the TwinMic beamline of Elettra—Sincrotrone Trieste (Trieste, Italy) [29]. The TwinMic microscope was operated in scanning transmission mode (STXM), where the specimen is raster-scanned across a microprobe delivered by a zone plate diffractive optic. The transmitted photons are collected through an X-ray-visible light converting system by a fast readout CCD camera providing absorption and differential phase contrast images [30]. Simultaneously, the XRF emitted by the sample is collected by eight SDDs located symmetrically in front of the sample [31,32]. The sample plane is located perpendicularly to the beam axis and the measurements are conducted at room temperature and in high vacuum condition. For the present experiment, an incident energy of 1.5 keV was chosen for optimal excitation of Na and Mg, allowing the detection of O as well. The zone plate optic (600 µm in diameter with an outermost zone width of 50 nm and a central stop of 140 µm) was focusing the incident X-ray photons on the sample on a spot size of 200 nm to 1 µm in diameter, according to the analysed sample and the incident beam intensity available during the experiments. The collected XRF spectra were deconvolved with the PyMCA software package [33]. For endometrial cells, at least three areas were mapped per fixation type, for a total of six analysed cells/fixation method, while for the HEK-293 cell line, at least four cells were mapped per fixation protocols. As far as for the sperm cells, for each of the three donors, a range of six to ten were analysed for each type of fixation.

The presented XRF counts and the relative MFC were calculated as average values on the whole cell.

A comparison about the microscopies used in the manuscript and a summary of a few experimental details are given in Appendix A. Please note that in the table we consider generic applications of those microscopies, but we acknowledge that special ones exist as well which may have quite different specifications (i.e., HS-AFM, high energy XRF, fly scan STXM, no-contact AFM, etc.).

### 4.4. Statistical Analysis

Data are expressed as means ± standard deviation or min-to-max box plots. One- or two-way ANOVA with multiple comparisons tests were carried out for statistical comparisons among the different fixation methods. Descriptive analysis was performed to calculate coefficients of variation (CV). Differences with a *p* < 0.05 were considered statistically significant. Data analyses were developed on GraphPad Prism 8.4.2 (GraphPad Software Inc., La Jolla, CA, USA). 

## Figures and Tables

**Figure 1 molecules-28-01992-f001:**
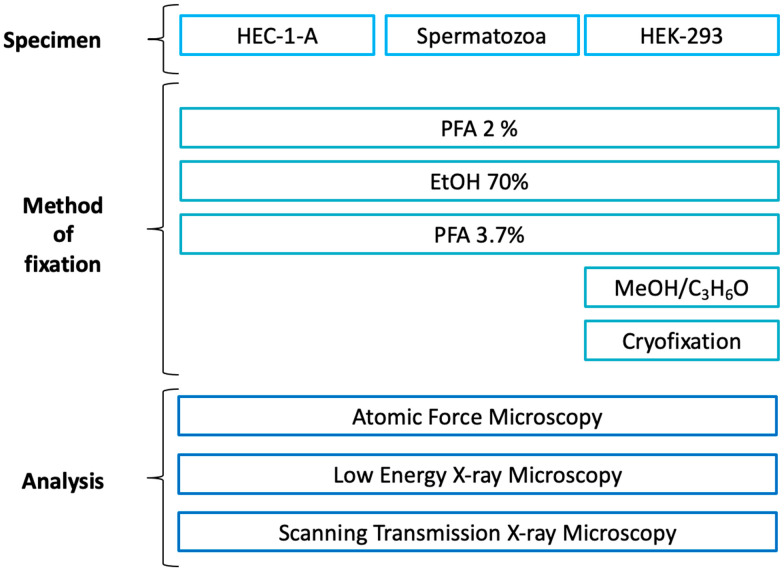
Experimental plan of the experiment indicating the fixation protocols used for each type of sample. Paraformaldehyde (PFA); Ethanol (EtOH); Methanol (MeOH); Acetone (C_3_H_6_O).

**Figure 2 molecules-28-01992-f002:**
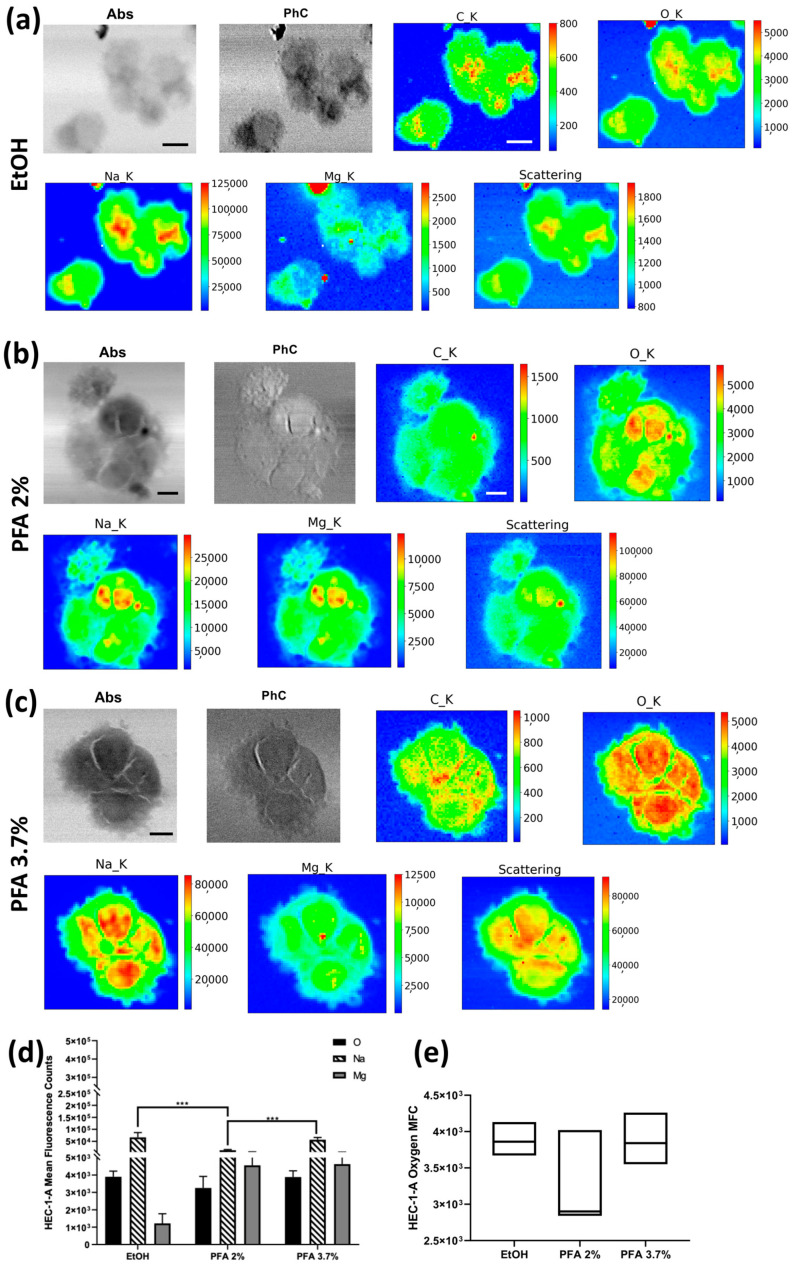
STXM and XRF analysis of the HEC-1-A cell line. Absorption (Abs) and differential phase contrast (PhC) images, alongside with O, Na, Mg, and Scattering maps of HEC-1A endometrial cells fixed with three different methods: (**a**) EtOH 70% (mapped area 60 × 52 µm^2^), (**b**) PFA 2% (mapped area 68 × 68 µm^2^), and (**c**) PFA 3.7% (mapped area 52 × 52 µm^2^). Scale bars are 10 µm. (**d**) Normalised average XRF counts of O, Na, and Mg in HEC-1-A adenocarcinoma endometrial cells fixed with EtOH 70%, PFA 2%, and 3.7% evaluated on at least 6 cells/fixation method. (**e**) Min-to-max box plot comparing oxygen MFC as obtained by descriptive statistical analysis used to determine the coefficient of variation. A two-way ANOVA with multiple comparisons test was performed to test the statistical difference among the fixation methods. *** *p* < 0.001. The STXM images (Abs and PhC) and XRF images were acquired at 1500 eV excitation energy with a step size of 200 nm and 800 nm, respectively, and an acquisition time of 10 ms/pixel and 6 s/pixel, respectively.

**Figure 3 molecules-28-01992-f003:**
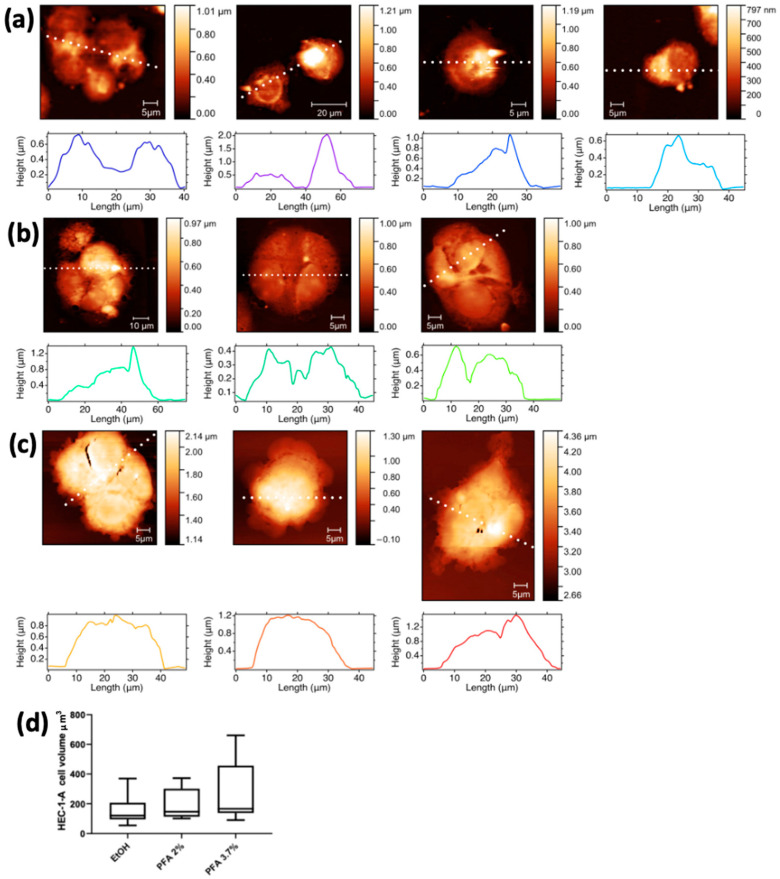
AFM images with corresponding surface profiles of the HEC-1A endometrial cells. The first column shows the same cells depicted in Figure 2, fixed with different methods: (**a**) EtOH, (**b**) PFA 2%, and (**c**) PFA 3.7%. The other pictures of panels a-c show additional cells prepared with the same three fixation methods. (**d**) Min-to-max box plot of HEC-1-A cell volume (µm^3^).

**Figure 4 molecules-28-01992-f004:**
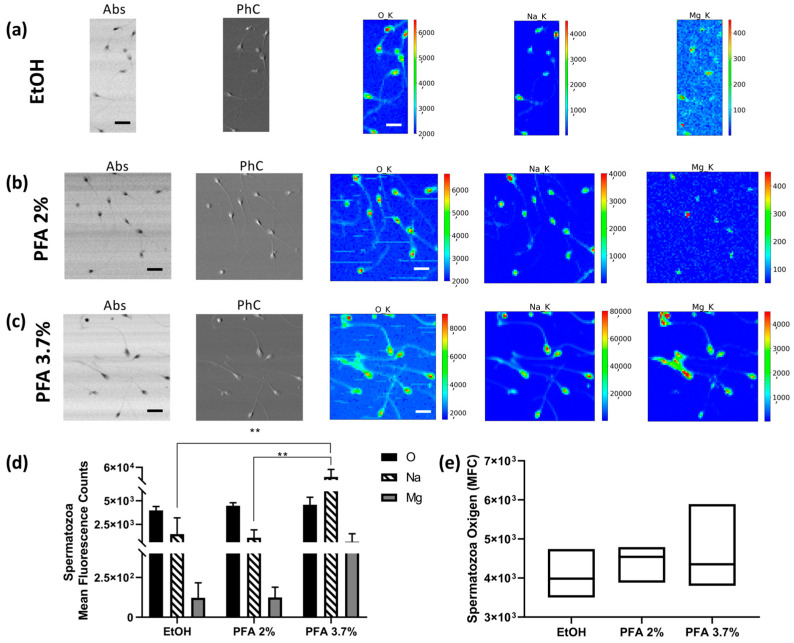
STXM and XRF analysis of human spermatozoa. Absorption (Abs) and phase contrast (PhC) images, together with O, Na and Mg maps of spermatozoa fixed with different methods: (**a**) EtOH 70% (mapped area 30 × 75 µm^2^), (**b**) PFA 2% (mapped area 70 × 70 µm^2^), and (**c**) PFA 3.7% (mapped area 70 × 70 µm^2^). Scale bars are 10 µm. (**d**) Normalised average XRF counts of O, Na, and Mg in spermatozoa fixed with EtOH 70%, PFA 2%, and 3.7% evaluated on at least 6–10 cells/patient/fixation method. (**e**) Min-to-max box plot comparing oxygen MFC as obtained by descriptive statistical analysis used to determine the coefficient of variation. A two-way ANOVA with multiple comparisons test was performed to test the statistical difference among the fixation methods. ** *p* < 0.005. The STXM images (Abs and PhC) and XRF images were acquired at 1500 eV excitation energy with a step size of 400 nm and 1 μm, respectively, and an acquisition time of 5 ms/pixel and 10 s/pixel, respectively.

**Figure 5 molecules-28-01992-f005:**
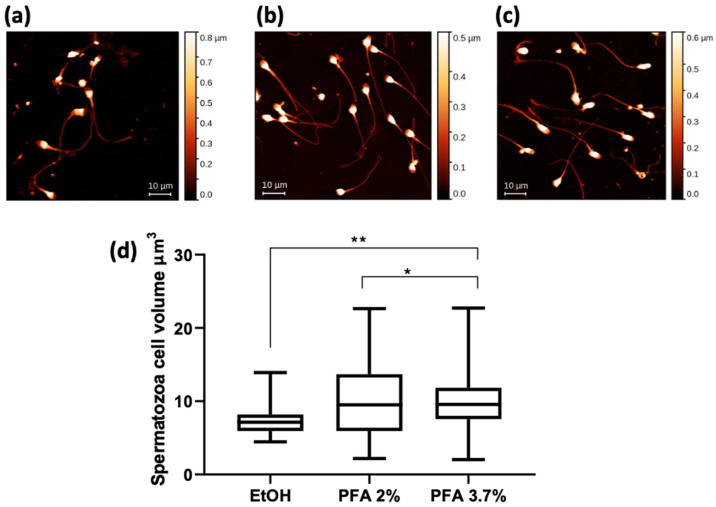
AFM images of human spermatozoa fixed with different methods: (**a**) EtOH, (**b**) PFA 2%, and (**c**) PFA 3.7%. Scale bar is 10 µm. (**d**) Spermatozoa cell volume data expressed as a min-to-max box plot. A one-way ANOVA with multiple comparisons test was performed to test the statistical difference among the mean volumes obtained after fixations. ** *p* < 0.005 * *p* < 0.05.

**Figure 6 molecules-28-01992-f006:**
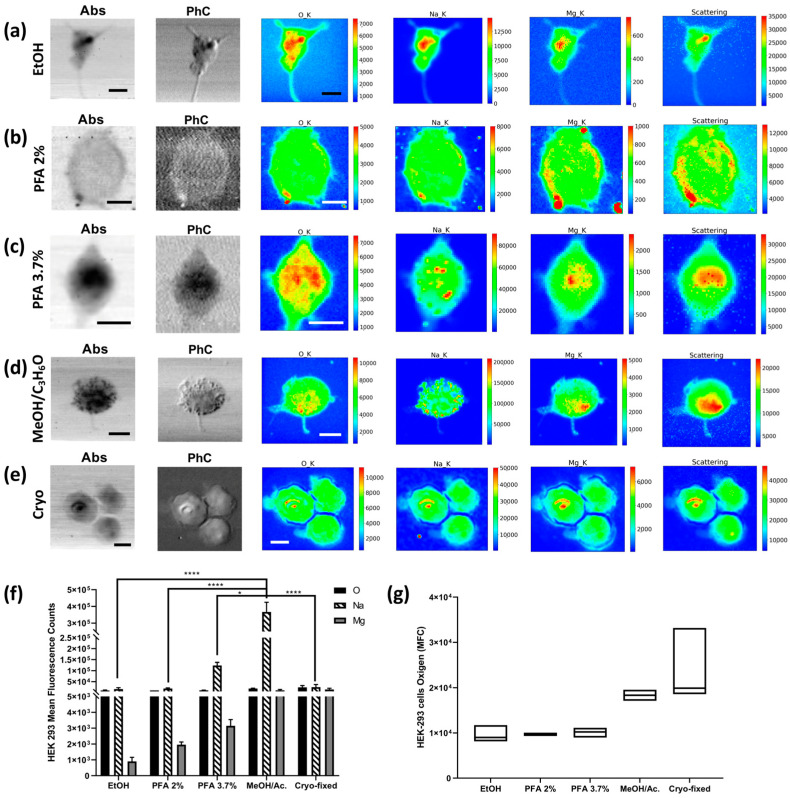
STXM and XRF analysis of HEK-293 cells. Absorption (Abs) and phase contrast (PhC) images, together with O, Na, Mg, and Scattering maps of HEK-293 cells fixed with different methods: (**a**) EtOH 70% (mapped area 45 × 45 µm^2^), (**b**) PFA 2% (mapped area 34 × 34 µm^2^), (**c**) PFA 3.7% (mapped area 25 × 30 µm^2^), (**d**) 1:1 MeOH/C_3_H_6_O (mapped area 40 × 40 µm^2^), and (**e**) cryofixed (mapped area 50 × 50 µm^2^). Scale bars are 10 µm. (**f**) Normalised average X-ray fluorescence counts of O, Na, and Mg in HEK-293. A two-way ANOVA with multiple comparisons test was performed on at least on 4 cells/fixation method to test the statistical difference among the fixation methods. * *p* < 0.05, **** *p* < 0.0005. (**g**) Min-to-max box plot comparing oxygen MFC as obtained by descriptive statistical analysis used to determine the coefficient of variation. The STXM images (Abs and PhC) and XRF images were acquired at 1500 eV excitation energy with a step size of 500 nm and an acquisition time of 33 ms/pixel and 6 s/pixel, respectively.

**Figure 7 molecules-28-01992-f007:**
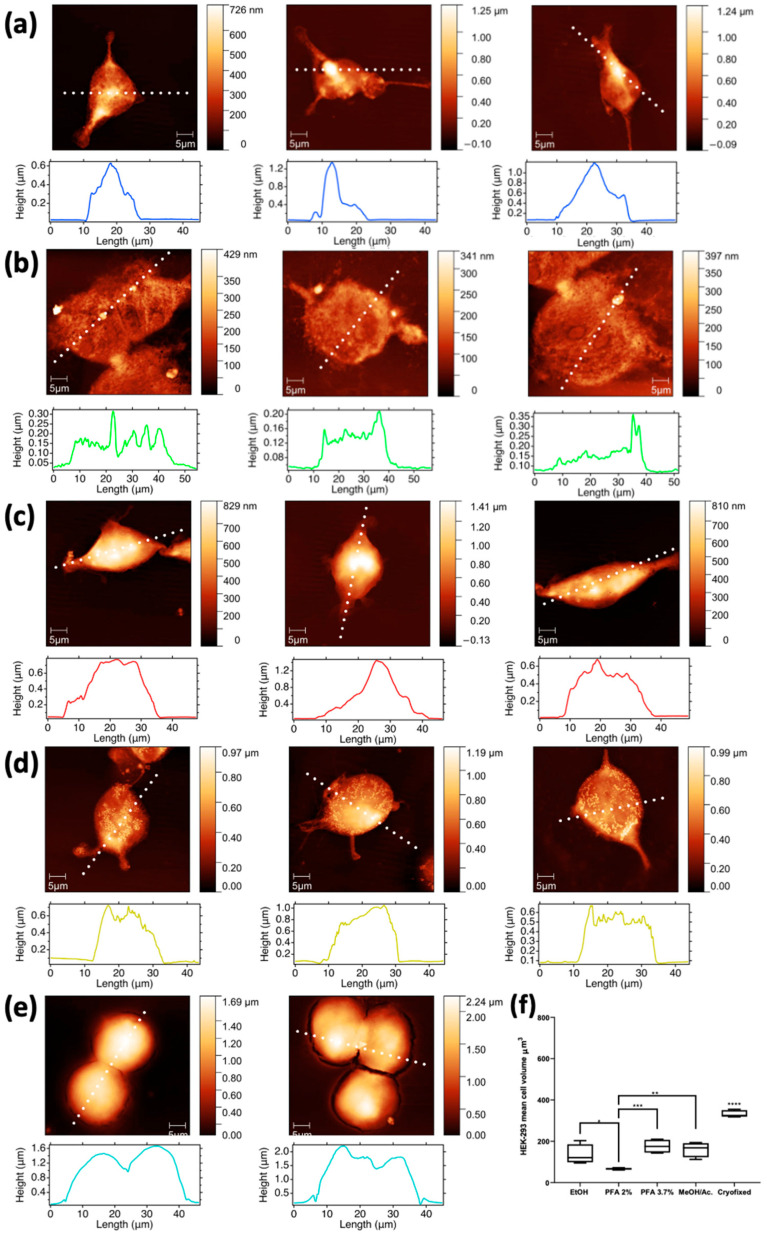
AFM images with corresponding surface profiles collected on a selection of HEK-293 cells fixed with the different methods: (**a**) EtOH 70%, (**b**) PFA 2%, (**c**) PFA 3.7%, (**d**) 1:1 MeOH/C_3_H_6_O, and (**e**) cryofixed. Scale bar is 5 µm. Cell volume calculated from the AFM images collected on a selection of HEK-293 cells fixed with ethanol and PFA 3.7% and compared with the rest of the applied fixation methods (**f**). The second column shows the same cells depicted in Figure 6 (where a, b and e are rotated of 90 degrees left, 90 degrees right and 180 degrees respectively compared to Figure 6), while the other columns depict additional cells prepared with the same five fixation methods. * *p* < 0.05, ** *p* < 0.005, *** *p* < 0.001, **** *p* < 0.0005.

**Table 1 molecules-28-01992-t001:** Summary of the main analytical constraints of the fixation methods under study for synchrotron X-ray microscopy.

Fixation Methods	Morphology Preservation	Artefacts	Elemental Content Preservation	Repeatability	Equipment Requirements
**PFA 2%**	Low	Yes	No	Low	Si_3_N_4_/SiC membrane
**EtOH 70%**	No	Low	No	High	Si_3_N_4_/SiC membrane
**PFA 3.7%**	Yes	No	Yes	High	Si_3_N_4_/SiC membrane
**MeOH/C_3_H_6_O**	No	Yes	No	High	Si_3_N_4_/SiC membrane
**Cryofixation**	Yes	No	Yes	Low	Si_3_N_4_/SiC membrane Vacuum pump Cryogen gas Liquid nitrogen

## Data Availability

The datasets generated and/or analysed during the current study are published as FAIR and open on a suitable data repository in Elettra Sincrotrone Trieste [34]. For any additional information, you may contact the corresponding author.

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
