# Peer review of "Impact of Sample Preparation Methods on Single-Cell X-ray Microscopy and Light Elemental Analysis Evaluated by Combined Low Energy X-ray Fluorescence, STXM and AFM"

_molecules, 2023, doi:10.3390/molecules28041992_

Round 1

Reviewer 1 Report (Previous Reviewer 2)

The problem addressed in the presented reseach is interesting, however the results shown in the paper exhibit poor statistical strenght due to small amount of samples analyzed. In the reviewers opinion the design of the research was inapropriate. I support my decision that due to above reason this paper does not diserve to be published in Molecules.

Author Response

We are well aware of the constraints of the study regarding the statistics. The small amount of analysed samples is intrinsically due to the nature of the study, which is synchrotron-based.

Synchrotron access is subjected to a peer review process carried out every 6 months and thus is not always guaranteed. Moreover XRF below 2 keV has very low efficiency, due to the low fluorescence yield in this energy range. Both these factors imply that only a limited number of samples can be analysed and long acquisition times are required.

To note, the studies available in literature suffer from the same limitation [i.e. refs 15 and 16: Perrin et al and Jin et al], however they still contain important valuable information and insights for the X-ray microscopy community. We hope that our paper as well will raise attention and interest on these, even dictate future needs.

In our scientific project we considered 3 different cell lines prepared with at least 3 fixation protocols to also consider variability among different types of samples/cell lines. Indeed, even if opinable, we favored testing different cell types rather than a single one with increased statistics.

Reviewer 2 Report (New Reviewer)

The paper outlines important information about LEXRF cell preparation, where in many cases cyrofixation is not viable, I recommend publication with some minor caveats. 

1. Freeze dried specimens were not included. I appreciate the difficulty in obtaining extra data, however I think the manuscript could be greatly improved by including some discussion, with reference to other studies by the Authors and other researchers.

2. The authors recommend PFA at 3.7% in the absence of cryofixation. However it does not appear as clear as this, and Fig 6 (and Figs 2 and 4 to a lesser extent) suggests that all chemical fixatives disturb some aspects of the measurement. The results suggest that armed with the knowledge of this study, researchers can choose their preferred method of cell preservation given the aims of the study and equipment available. A short table with yes/no options in the discussion would be very handy and increase the usability of this work. This would be especially true if freeze dried preparation could be included in such a table.

Author Response

The paper outlines important information about LEXRF cell preparation, where in many cases cyrofixation is not viable, I recommend publication with some minor caveats. 

We thank the reviewer for the positive feedback and the suggestions which allowed us to improve our manuscript.

  1. Freeze dried specimens were not included. I appreciate the difficulty in obtaining extra data, however I think the manuscript could be greatly improved by including some discussion, with reference to other studies by the Authors and other researchers.

We thank the reviewer for his/her comment. We apologize for the misunderstanding, indeed we were not explicit in our description. Freeze dried specimens are included in the samples analysed in our experiments. The cryofixed cells were in fact freeze-dried after plunge. We make it now more explicit in the Materials and Methods, beside better citing literature (JIn et al)..

  1. The authors recommend PFA at 3.7% in the absence of cryofixation. However it does not appear as clear as this, and Fig 6 (and Figs 2 and 4 to a lesser extent) suggests that all chemical fixatives disturb some aspects of the measurement. The results suggest that armed with the knowledge of this study, researchers can choose their preferred method of cell preservation given the aims of the study and equipment available. A short table with yes/no options in the discussion would be very handy and increase the usability of this work. This would be especially true if freeze dried preparation could be included in such a table.

We thank the reviewer for this remark. We included a table as suggested and we added a more explicit comment in the Discussion session.

Reviewer 3 Report (New Reviewer)

This manuscript describes a very important issue regarding the single-cell microscopy. I found this paper was well organized and very informative. To be more comprehensive, it is suggested that the authors should provide additional information :

1)the spatial resolution of STXM needs to be addressed. 

2) Comparison of detection depth and sensitivity of the each techniques., i.e. Low energy X-ray fluorescence), STXM  and AFM in a Table.

3) Radiation damage issues were mentioned but not addressed clearly. Is it possible to qualititatively estimate the effects from strong to low for the three discussed techniques?

Author Response

This manuscript describes a very important issue regarding the single-cell microscopy. I found this paper was well organized and very informative.

We thank the reviewer for the positive feedback and the suggestions which allowed us to improve our manuscript and to make it clearer for the readers.

To be more comprehensive, it is suggested that the authors should provide additional information :

1) the spatial resolution of STXM needs to be addressed. 

Thank you for spotting this, we added the related information in the figures’ captions and in the Materials and Methods section.

2) Comparison of detection depth and sensitivity of the each techniques., i.e. Low energy X-ray fluorescence), STXM  and AFM in a Table.

We thank the reviewer for this remark, however it is not easy to compare the 3 techniques in all the above mentioned specifics as AFM is based on very different physical phenomena compared to STXM and LEXRF albeit they are all scanning techniques. A detailed analysis could be indeed interesting but will be lengthy. Nevertheless in the Supporting Information Material we added a Table (Table S5, see here below) that summarises selected characteristics of each technique that we consider relevant to the topic of our manuscript.

In contrast to STXM (transmission) and XRF (emission),  AFM in most practical applications is only surface technique thus there is no  detection depth and also it does not provide chemical information. It can provide morphological and topographical information though. In some specific experimental setup AFM can also provide some mechanical information though Young’s modulus measurement (DOI: 10.1039/D0NR02314K). Please note, that we consider generic application of those microscopies but we acknowledge that special ones exist that may have quite different specification (ie. HS-AFM, high energy XRF, fly scan STXM,no-contact AFM, etc ).

Technique

Attenuation length @ 1500eV

Chemical Sensitivity

Lateral Resolution

Vertical resolution

Measurement time per pixel

Sample environment

AFM

NA

NA

80-100nm

0.5 nm

20-40 ms

Air

STXM

5um

NA

200-1000nm

NA

10-20ms

Vacuum

LEXRF

0.2um for O, 1um for Na, 3um for Mg emissions

ppm for chemical elements

200-1000nm but for signal reasons >400nm

NA

1-5 s

Vacuum

The detection depth has been indicated by evaluating the attenuation length, which is the distance from the surface into a material where the X-ray beam intensity has decreased to a value of 1/e (~ 40%) of the incident beam intensity.

While for STXM, which is based on the collection of the transmitted photons, the incoming photons of 1500eV are transmitted with a great probability, for LEXRF, which is based on the collection of the emitted photons excited from the incoming photons inside the sample, the detection depth varies according the the emission energy.

For STXM and LEXRF we report the current available options at TwinMic beamline. Other synchrotron beamline may have different parameters.

3) Radiation damage issues were mentioned but not addressed clearly. Is it possible to qualititatively estimate the effects from strong to low for the three discussed techniques?

AFM does not induce any radiation damage on the biological sample, as it does not rely on the use of ionising radiation.

On the other hand, X-ray sources cause radiation damage to the extent that it is proportional to the number of photons (dose) delivered on the specimen. The radiation damage induced by STXM and LEXRF is of the same nature, as they use the same radiation source (soft X-rays); the difference relies on the exposure time/pixel used for STXM and LEXRF, that is milliseconds compared to seconds respectively. From our previous studies [Gianoncelli et al Soft X-Ray Microscopy Radiation Damage On Fixed Cells Investigated With Synchrotron Scientific Reports, Vol. 5, 10250 (2015) doi: 10.1038/srep10250] we can say the STXM imaging usually does not induce a strong radiation damage while LEXRF does. So to summarize, concerning the radiation damage, AFM No, STXM low and LEXRF strong.

We thank the reviewer for the valuable suggestion worth studying in a subsequent paper.

Round 2

Reviewer 1 Report (Previous Reviewer 2)

Publish as is.

This manuscript is a resubmission of an earlier submission. The following is a list of the peer review reports and author responses from that submission.

Round 1

Reviewer 1 Report

The authors compared different sample preparation techniques for X-ray fluorescence analysis (XRF) at synchrotron radiation facilities. The analyses were made using not only XRF but transmission and phase contrast imaging as well as AFM. The paper presents useful information for those who work with SR-XRF. There are some issues that need attention, particularly the limitations of the study should be made clear. Comparison of the techniques is very useful but a caveat is that these have to be made on different cells. Did the authors considered comparing the ratio of elements, under the assumption that they are the same among different cells, or normalizing fluorescence with cell volume or transmittance?

Line 62, Conditions of the XRF measurement assumed in this paper should be explained better. It is possible to apply XRF on cells without fixation. Transition metals do not require XRF measurements under vacuum (drying is as worrisome as fixation concerning preservation of structure). Use of SiN is not mandatory, thin films can be used. All these are not required for XRF but for a specific type of a beamline.

Line 89, “Mild x-ray range” should be more specific.

Line 101, Some explanation is necessary for the measurement of cryofixed samples. In some cases, frozen cells are investigated in a frozen hydrated state under cooling. Again, drying is an important issue.

Line 139, Phase-contrast imaging, which appears first time here, needs some explanation (maybe in Introduction).

Line 157, Could you discuss oxygen content in terms of the drying process? Much oxygen is lost with water.

Line 175, Is it possible to confirm that the cells are still attached to the SiN membrane?

Lines 237 and 244, “methanol/ C3H6O e” typo?

Lines 310-320 may need editing.

Lines 394 and 207, “Donors” may be more adequate than “Patient.”

Line 420, Were the AFM measurements conducted on the cells prepared as in section 4.1?

Line 435, Were the samples set perpendicular to the X-ray beam? and under vacuum at RT?.

Figure S1, Can the precipitation of Na be a crystal of salt? This can be clarified by measuring XRF of Cl.

Reviewer 2 Report

General remarks:
1.    The presented case study is interesting escpecially that light elements were not studied in similar works, however, the presented data clearly shows that the cells of each kind exhibit extremely large diversity of shapes and sizes. In that case the sample population should be bigger to get good statistics, especially if you want to compare cells volumes.
2.    Why HEC-1-A and spermatozoa were not cryofixed and/or treated with MeOH/C3H6O? Cryofixed samples as the least affected by the chemical treatment should be a reference in all the cases.
3.    The fluorescence counts doesn’t seems to be an appropriate signal for quantitative comparison of elemental concentration of the samples which are very heterogenous in terms of shape, morphology and thickness. This signal is strongly influenced by matrix effects, especially for oxygen (variable topography of the samples) – the Authors should address that problem, especially that some of the Authors have an experience in quantification of such samples, and supportive AFM data are available.
4.    It is unclear how many cells of each kind were analyzed by XRFM? I guess that this is the “number of values” given in tables S1-S3. It should be clarified.
5.    Tables S1 – S3. The calculation of estimated standard deviation and standard deviation of the mean value should be done for at least 10 values, whereas the number of values is not greater than 8 (typically 3). This is the basic knowledge on the measurement uncertainty. For such small number of samples other approaches should be used (see some fundamental works like the work of Dean & Dixon 1951).
6.    The PhC image of HEC-1-A after EtOH treatment looks awkward – it shows no edges. The same applies for HEK-293 after PFA. More comment on those particular images should be given.
7.    All PhC, Abs and XRF data for all samples measured should be included in the supplementary materials.
8.    How does the MFC were calculated for a single cell from the XRF map data? Please describe it in details.
9.    The conclusions of AFM examination of HEC-1-A are unclear and seems to be not poorly supported by the measurement data. Page 6 line 181 – what does it mean that “the membrane edges tend to become thinner”? How this information was evaluated from AFM mapping? Page 6 line 183-184: those three values are equal within their uncertainties! You are not allowed to say that one is bigger than other. This statement is simply not supported by data.
10.    The purity of the reagents used for fixation should be investigated.
11.    What was the time of acquisition and step size for Abs/Phc/XRF mapping?
12.    Table S3 – PFA 2% - are there really two identical values?
13.    Did the Authors checked if the data fulfill the appropriate requirements before using ANOVA?

Editorial errors:
1.    The editing of the text is messy in some places. Please read it carefully and correct.
2.    The language should be improved – there is a lot of minor errors.
3.    Figure 2/4/6 captions – the values in brackets are not described (map area I guess).
4.    Figure 3 – the scales are not readable.
5.    The information about fixation protocol should be given directly on the figure 2/4/6 – it would be much easier to read this image since the captions contains a lot of information.
6.    Figure 4 – why the Abs and PhC images are not showing the same area? The PhC at b) corresponds to Abs at c) after up and down mirroring. The same for PhC at c) and Abs at b) and the same for PhC and Abs for a).

Round 2

Reviewer 1 Report

Thank you for clarifications and corrections.

Reviewer 2 Report

The manuscript has been partially improved however the main drawbacks still holds:
1. The sample population is to small (or the diversity of cell sizes and shapes is to large) to enable any reasonable conclusion (especially for AFM measurements). The fact that the XRF and AFM measurement are not high-throughput methods means that the scientific project was not properly designed in terms of methodology.
2. If the authors want to compare the elemental content of the samples the quantification is not beyond the scope of this paper.
3. The uncertainties were not corrected in the text (changes inly in the supplementary material).
4. Using ANOVA for such small populations is improper (it is not possible to check if the distribution is normal with 3 samples).
The aim of the Molecules journal is to publish “cutting-edge” research. In the reviewer’s opinion the results shown is far from being “cutting-edge”. I maintain the decision that the material is not suitable to be published in Molecules.